# Direct bandgap emission from strain-doped germanium

Lin-Ding Yuan [1,3], Shu-Shen Li [1,2] & Jun-Wei Luo [1,2] ✉

Germanium (Ge) is an attractive material for Silicon (Si) compatible optoelectronics, but the nature of its indirect bandgap renders it an inefficient light emitter. Drawing inspiration from the significant expansion of Ge volume upon lithiation as a Lithium (Li) ion battery anode, here, we propose incorporating Li atoms into the Ge to cause lattice expansion to achieve the desired tensile strain for a transition from an indirect to a direct bandgap. Our first-principles calculations show that a minimal amount of 3 at.% Li can convert Ge from an indirect to a direct bandgap to possess a dipole transition matrix element comparable to that of typical direct bandgap semiconductors. To enhance compatibility with Si Complementary-Metal-Oxide-Semiconductors (CMOS) technology, we additionally suggest implanting noble gas atoms instead of Li atoms. We also demonstrate the tunability of the direct-bandgap emission wavelength through the manipulation of dopant concentration, enabling coverage of the mid-infrared to far-infrared spectrum. This Ge-based light-emitting approach presents exciting prospects for surpassing the physical limitations of Si technology in the field of photonics and calls for experimental proof-of-concept studies.

Despite Silicon being the primary semiconductor for electronics, the nature of its indirect bandgap renders it an inefficient light emitter and, thus, hinders its use for photonic applications[1-4]. A variety of approaches[5,6] has been explored since the 1980s to obtain efficient light emission from group IV semiconductors. Approaches like porous Si[7,8], Si nanostructure[9-13], Silicon-Germanium (SiGe) superlattices[14], Erbium (Er) doped Si-rich materials[15,16], and allotropes of Si[17-19] and Ge[20,21], strained Ge on Si substrate[22,23], GeSn alloy[22,23] all to some extend has its limitations. The pursuit of CMOS-compatible light emission from group-IV materials remains a grand challenge[20].

After the successful introduction of Ge in the source/drain regions in the 90 nm logic technology node to create a uniaxial compressive strain on the Si channel[24], the integration of Ge in Si has steadily evolved, and Ge is now recognized as a "standard" CMOS material[25]. Specifically, Ge has been demonstrated to realize the ultrafast (a 3 dB bandwidth of 265 GHz) Si-integrated waveguide-photodetectors in compatibility with silicon photonics and CMOS

fabrication[26,27]. Thanks to Ge's almost direct bandgap---the direct bandgap $E_g^\Gamma$ is only about 0.14 eV above its fundamental indirect bandgap $E_g^L$, theoretical work[28] predicted that a 2% biaxial tensile strain is sufficient to transform Ge from an indirect bandgap to a direct bandgap. However, the introduction of such a level of biaxial tensile strain in Ge remains inaccessible, especially, in a CMOS-compatible way. Strain achieved through the traditional strategy of heteroepitaxy Ge-on-Si is compressive due to Ge having a lattice constant 4.2% larger than Si[29]. The tensile strain created by thermal annealing of Ge files grown on Si is limited to 0.25%[30]. A variety of mechanical approaches[31-33] has been used to exert the required 2% tensile strain on Ge, but are not compatible with CMOS technology. Furthermore, considering Sn possesses a much smaller $E_g^\Gamma$ than $E_g^L$, alloying Ge with Sn was predicted to remarkably reduce the bandgap at the $\Gamma$ point and achieve an indirect-to-direct transition as increasing Sn composition[23,34-36]. However, the low equilibrium solubility of Sn in Ge (<1%) and compressive strain induced by lattice

[1]State Key Laboratory of Superlattices and Microstructures, Institute of Semiconductors, Chinese Academy of Sciences, Beijing 100083, China. [2]Center of Materials Science and Optoelectronics Engineering, University of Chinese Academy of Sciences, Beijing 100049, China. [3]Present address: Department of Materials Science and Engineering, Northwestern University, Evanston, IL 60208, USA. ✉e-mail: jwluo@semi.ac.cn

mismatch substrate pose a big challenge towards on-chip high-performance lasers[35,37,38].

In this letter, we propose a CMOS-compatible route to introduce the desired tensile strain to realize the direct bandgap emission from Ge. Given that Ge undergoes a large volume expansion during lithiation[39] as a Li-ion battery anode, here, we suggest implanting Li atoms into Ge (Fig. 1a) to expand the lattice in order to produce the required tensile strain for an indirect-to-direct bandgap transition. By conducting the first-principles calculations, we show that a small amount of Li atoms (~3%) can convert Ge from an indirect bandgap to a direct bandgap with a dipole matrix element comparable to direct bandgap III-V semiconductors. It sharply contrasts with the direct bandgap hexagonal Ge, which was predicted to exhibit a very weak optical dipole transition due to its quasi-direct bandgap[40]. The recently observed finite emission in hexagonal Ge[20] may be a consequence of extrinsic effects[41]. For better compatibility with the Si CMOS technology, we further suggest using noble gas atoms instead of Li atoms. We find a lower critical concentration for the indirect-to-direct bandgap transition for a larger atom size: 7.8, 1.6, 0.8, and 0.8 at.% for He, Ne, Ar, and Kr, respectively. We show that the direct-bandgap emission wavelength is tunable by increasing dopant concentration to cover the mid-infrared to far-infrared spectrum -- a leading candidate for the next optical communication window[42–44].

## Results

### Indirect-to-direct bandgap transition in Li-doped Ge

Figure 1 shows the hybrid functional (HSE06)[45] calculated effective band structure (EBS) for the Li-doped Ge (Ge:Li) at Li concentration 3.1 at.% (units in relative atomic concentration at.%) in comparison with the band structure of the parent Ge. We adopt the experimental lattice parameter for crystalline Ge at 300 K[46], and scaled lattice constants for doped Ge systems where no experimental data is available. We use a special quasi-random structure approach (SQS)[47] within a 32 to 64 atoms supercell to generate the Li distribution by assuming Li is randomly distributed at the tetrahedral interstitial sites in Ge (shown in Fig. 1a). Details about the structure models and justification for Li to stay at tetrahedral interstitial sites can be found in the Method section. The transition from the valence band maximum (VBM), which is located at the Γ point, to the Γ-valley of the conduction band is named the direct bandgap $E_g^\Gamma$, and the transition from the VBM to L-valley is

indirect bandgap $E_g^L$. Figure 1b shows that at Li concentration 3.1 at.% the Γ-valley of Ge:Li is at the same level as the L-valley, indicating that the implantation of Li atoms has yielded an indirect-to-direct bandgap transition for Ge.

Figure 1c (upper panel) shows the predicted energies of $E_g^\Gamma$ and $E_g^L$ of Ge:Li systems as a function of Li concentration. One can see that both $E_g^\Gamma$ and $E_g^L$ get lower in energy as increasing Li concentration but $E_g^\Gamma$ level drops faster than $E_g^L$ level. It is thus expected that, by increasing Li concentration, Ge should have a transition from indirect bandgap to direct bandgap, which occurs when Li concentration achieves around 3.1 at.%, evidenced by the crossing of $E_g^\Gamma$ and $E_g^L$ at around 0.6eV. Further increasing Li concentration, the fundamental bandgap becomes direct and gets smaller linearly, giving rise to direct bandgap emission with wavelength covering the mid- to far-infrared spectral region. The corresponding fundamental bandgap dipole matrix element $E_p$ between the conduction band minimum (CBM) and VBM is shown in the lower panel of Fig. 1c. One can see that $E_p$ is zero for pure Ge owing to its indirect bandgap nature. The implantation of Li atoms makes $E_p$ finite with its magnitude growing linearly with increasing Li concentration. At 3.1 at.% Li concentration, $E_p = 10.0$ eV is about half the value to that of the direct-bandgap InP ($E_p = 20.7$eV)[48]. which is consistent with the observation that the CBM state contains 58.3% of Γ-Bloch character a full admixture of Γ-valley and L-valley (See Supplementary Notes 1 and Supplementary Fig. 1). When Li concentration exceeds 6.2 at.%, $E_p$ reaches a steady value comparable to the direct-bandgap InP, indicating the achievement of a truly direct bandgap with a high-efficiency light emission. This is in sharp contrast to the quasi-direct bandgap obtained in nanostructured Si and hexagonal Ge[20] in which the bandgap dipole matrix element is weak with a finite strength arising from Γ-X mixing[13] or Γ-L mixing[40], respectively. Note that the L-valley of cubic Ge folds into the Γ-point in the hexagonal Ge and couples with the high-lying state derived from the Γ-valley of cubic Ge. This coupling repels the Γ-valley-derived state upward and L-valley-derived state downward, necessitating a larger hydrostatic tensile strain of approximately 5% to transform the quasi-direct bandgap into a direct bandgap in hexagonal Ge[41].

### Volume expansion as the main driving force

To verify whether the volume expansion is a leading factor driving the indirect-to-direct bandgap transition, we evaluate the effective strain

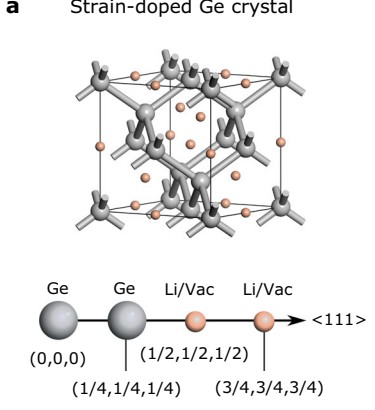

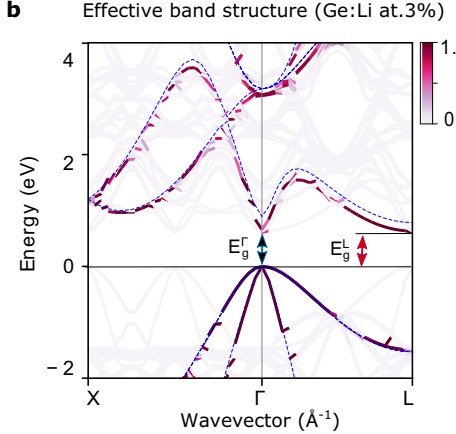

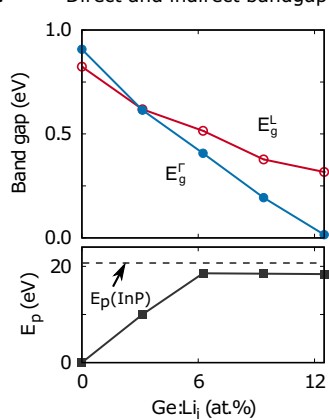

**Fig. 1 | Indirect-to-direct bandgap transition in Ge achieved by implantation of Li atoms. a** Crystal structure of Li implanted Ge (Ge:Li) with all tetrahedral interstitial sites filled by Li atoms. **b** Effective band structure of Ge:Li with a Li concentration of 3 at.% (colored solid lines) in comparison with the band structure of parent Ge (dashed blue lines). They are aligned in view of the same valence band maximum (VBM). The darkness of the solid lines of the unfolded bands represents the weight of the spectrum function (ranging from 0 to 1). It tells the probability of

an electron at wavevector $\boldsymbol{k}$ with energy $E$. **c** Upper panel: The energies of direct bandgap $E_g^\Gamma$ and indirect bandgap $E_g^L$ of Ge:Li as a function of Li concentration, and Lower panel: corresponding fundamental bandgap dipole matrix element $E_p$ ($E_p = P^2/2m_0$ where $P = i\langle c|\hat{p}|v\rangle$ is the dipole matrix elements between the conduction band edge and valence band edge). The dashed line represents the dipole matrix element $E_p$ of direct bandgap InP.

induced by implanting Li atoms into the Ge lattice. Figure 2a shows that the magnitude of the effective tensile strain is linearly proportional to the Li concentration. Because $\Gamma$- and $L$-valley of the conduction band are the antibonding states in cubic semiconductors, a tensile strain elongating the bonds will pull down their energy levels. However, the $\Gamma$-valley drops faster than the $L$-valley since $\Gamma$-valley is a pure s-orbital and $L$-valley is an admixture of s- and p-orbitals[49]. Indeed, Fig. 2b shows that a stronger tensile strain pulls both $E_g^{\Gamma}$ and $E_g^{L}$ down but $E_g^{\Gamma}$ drops faster than $E_g^{L}$. It results in an indirect-to-direct bandgap transition in Ge as the tensile strain $\varepsilon$ exceeds 0.5%. Note that, here, the tensile strain is hydrostatic. In the case of biaxial strain enforced by a substrate, the required tensile strain for indirect-to-direct bandgap transition is predicted to be 1.5%, which is slightly smaller than the literature-reported value of 2.0%[28] due to we adopted the experimental lattice constant instead of the DFT value used in the literature. Figure 2a shows that the required 0.5% effective tensile strain for indirect-to-direct transition is attained in Li-doped Ge with a Li concentration at ~3.1 at.%, which is also the Li concentration predicted to undergo the indirect-to-direct transition in Li-doped Ge, as shown in Fig. 1c. Such excellent agreement demonstrates doping-induced tensile strain being the prime factor causing the indirect-to-direct bandgap transition in Li-doped Ge.

## Si$_x$Ge$_{1-x}$ substrate effect

So far, we have demonstrated the indirect-to-direct bandgap transition by incorporating Li atoms into free-standing Ge. We now consider the effect of biaxial strain on Ge:Li systems arising from a Si$_x$Ge$_{1-x}$ substrate. Figure 3 shows the evolution of energy levels of the direct and indirect bandgaps $E_g^{\Gamma}$ and $E_g^{L}$ as a function of Li concentration in the

Ge:Li system epitaxially grown on the Si$_x$Ge$_{1-x}$ substrates (x = 0, 0.2, 0.4). In comparison to the freestanding case, the pseudomorphic Ge:Li layer on a Ge substrate (x = 0) requires a larger Li concentration (6.2%) to achieve a direct bandgap. This biaxial strain effect induced by the Si$_x$Ge$_{1-x}$ substrate becomes more pronounced with a lower Ge composition. Figure 3c shows that when x = 0.4 the increase of Li concentration is unlikely to reverse the order of $E_g^{\Gamma}$ and $E_g^{L}$, and thus cannot transform the Ge:Li from indirect bandgap to direct bandgap. It implies that a Ge-rich Si$_x$Ge$_{1-x}$ substrate (<0.4) is necessary to obtain Li-doped direct bandgap Ge. Fortunately, the technology of high-quality relaxed Ge layers grown on the Si substrate is now mature[25,27,50].

## Indirect-to-direct bandgap transition in Noble gas atom doped Ge

Considering Li may be incompatible with current CMOS technology, we alternatively suggest using noble gas atoms to strain dope Ge. Unlike Li, noble gas atoms with closed-shell valence electron configuration are chemically inert and will not donate electrons to Ge; Thus, being an ideal dopant for strain doping. Figure 4 shows the evolution of energy levels of $E_g^{\Gamma}$ and $E_g^{L}$ as a function of doping concentration for He, Ne, Ar, and Kr dopants. It exhibits an indirect-to-direct bandgap transition occurring in all Ge:X (X = He, Ne, Ar, and Kr) systems, confirmed by the sharp growth of the fundamental bandgap dipole matrix elements. The critical doping concentration for the indirect-to-direct bandgap transition is smaller for a larger atom size: 7.8, 1.6, 0.8, and 0.8 at.% for He, Ne, Ar, and Kr with a bandgap of 0.73, 0.78, 0.69, and 0.63 eV, respectively. We should note that the predicted critical doping concentration may not be very precise given the largest supercell considered here contains only 128 atoms and the critical concentration 0.8 at.% corresponds to 1 atom in a 128 atoms-supercell. Nonetheless, the increase in the dopant concentration will lead to a continuous decrease in the magnitude of the achieved fundamental bandgap which covers the mid-infrared to far-infrared spectrum. Particularly, all systems can be engineered to direct-bandgap emission of the 2 μm (0.62 eV) wavelength, which is a leading candidate for the next optical communication window[42–44].

## Discussion

We note that the filling of tetrahedral interstitial sites by noble gas atoms has been earlier proposed theoretically[51,52] to convert the indirect bandgap of semiconductors, such as Si, GaP, and ZnP, to direct bandgap based on, however, a vastly different mechanism. In the earlier proposal, all or half of the tetrahedral interstitial sites have to be fully filled by noble gas atoms due to the indirect-to-direct transition is realized mainly by moving up the X-valley over the Γ-valley in view of the X-valley states possessing a high density at the tetrahedral interstitial sites and could be selectively shifted upwards by filling atoms via an electron-repelling effect[51,52]. In our proposed scheme, the electron-repelling effect is absent and only a small fraction of the interstitial sites are randomly filled, which renders it to be much easy to synthesize in CMOS-compatible technology.

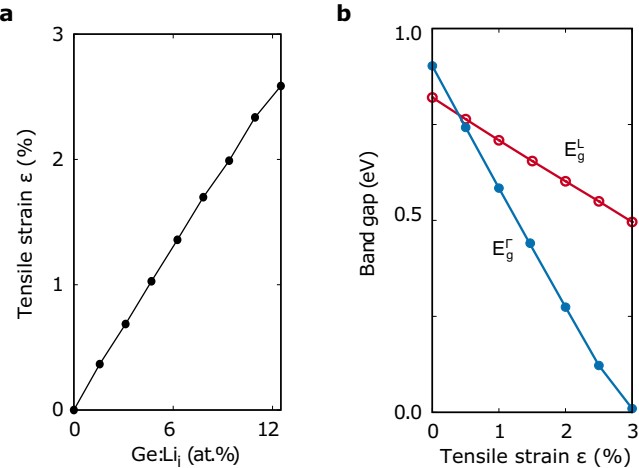

**Fig. 2 | Hydrostatic tensile strain induced indirect-to-direct bandgap transition in crystalline Ge. a** Effective tensile strain induced by implantation of Li atoms in Ge. **b** The effect of hydrostatic tensile strain on energies of direct bandgap $E_g^{\Gamma}$ and indirect bandgap $E_g^{L}$.

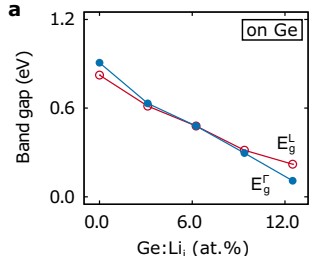
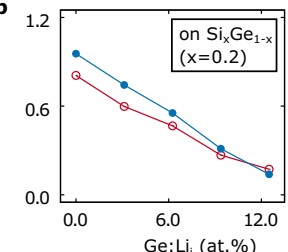
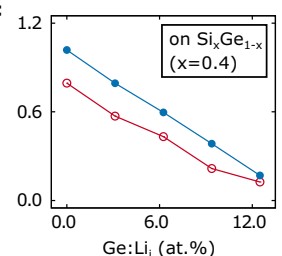

**Fig. 3 | Effect of biaxial compressive strain induced by Si$_x$Ge$_{1-x}$ substrate.** Direct bandgap $E_g^{\Gamma}$ and indirect bandgap $E_g^{L}$ as a function of Li atomic concentration in Ge:Li system on Si$_x$Ge$_{1-x}$ substrate for (**a**) x = 0; (**b**) x = 0.2; and (**c**) x = 0.4.

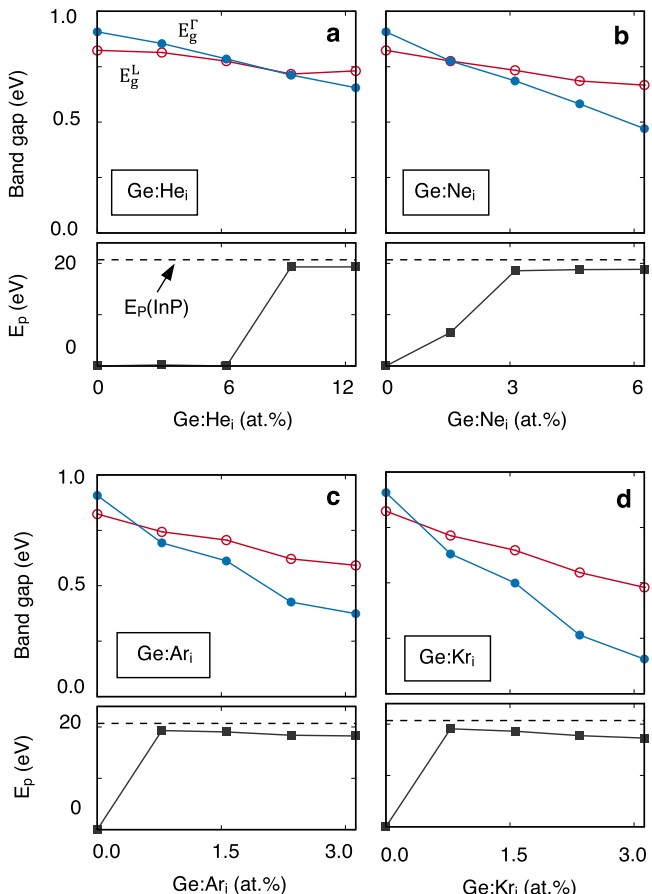

**Fig. 4 | Indirect-to-direct bandgap transition in Ge induced by the insertion of noble gas atoms.** The energies of direct bandgap $E_g^\Gamma$ and indirect bandgap $E_g^L$ (upper panel) of Ge:X as a function of X concentration and corresponding fundamental bandgap dipole matrix element $E_p$ ($E_p = P^2/2m_0$) (lower panel). The dashed line represents the dipole matrix element $E_p$ of direct bandgap InP. Inserted atoms X being (**a**) He, (**b**) Ne, (**c**) Ar, and (**d**) Kr.

Diffusion of Li into Ge followed by drift to compensate acceptor sites is a standard technology to fabricate the Ge:Li gamma-ray detectors developed in the 1960s[53]. Although the solubility of Li in Ge crystal falls within the range of $10^{17}$–$10^{18}$ cm$^{-3}$[54], the Li concentration as high as $1\times10^{21}$ cm$^{-3}$ (~9.0 at.%) has been reported in the surface layer of Si$_x$Ge$_{1-x}$ (x = 97.4 and 94.6%) crystals following a 10-minute diffusion at 320 °C measured directly by performing secondary mass ion spectrometry (SIMS)[55]. The results exhibit a clear decrease in Li concentration versus distance from the surface and higher Ge composition in Si$_x$Ge$_{1-x}$ crystals having a thicker high Li concentration surface layer with a higher peak concentration. It was attributed to the enhanced diffusion of lithium through interstitial positions by local strains created by Ge atoms. The observed low Li concentration in the Li-diffused crystalline Ge may exclude the possibility of Li-rich Li$_x$Ge phases[56], which occurs usually in the electrochemical lithiation of Ge[39]. To suppress the out-diffusion of Li atoms, we could deposit a highly compact oxide layer on top of the Ge:Li layer. The accumulation of Li atoms at the Ge surface and the wide bandgap of the deposition oxide layer may create a triangular-well confinement potential across the oxide/Ge:Li junction for both electrons and holes, which will benefit the lasing.

Doping far beyond the solubility limit can also be obtained by non-equilibrium doping techniques such as ion implantation followed by ultrafast annealing[57]. For instance, a saturation of interstitial lithium in the Si membrane at a concentration of about 10 at.% has been demonstrated by a direct-write lithiation of 35 nm thick crystalline Si membranes using a focused ion beam of Li[58]. The interstitial Li

impurities introduce no deep level within the Ge bandgap, but acts as a donor in Ge[59]. Because Li has one valence electron in its 2 s orbit, it shifts the Fermi level into the conduction band due to a hyper-doping, which may limit the light-emitting performance. But we can assess the transition from indirect to direct bandgap in Li-doped Ge from the photoluminescence spectroscopy since photo-excited carriers will relax to band edges in picosecond timescale and then radiatively recombine to generate photons. Therefore, it is still worth checking Li doped Ge experimentally, given that Li doping in Si and Ge is very accessible and has been widely applied in Li battery applications.

Unlike Li, noble gas atoms with closed-shell valence electron configuration are chemically inert and will not donate electrons to Ge; thus, more feasible for strain-doping induced indirect-to-direct bandgap transition in Ge. This is encouraged by a recent experiment[60] that has successfully used helium ion implantation to increase the out-of-plane lattice constant by 1% in epitaxial La$_{0.7}$Sr$_{0.3}$MnO$_3$ thin films. A maximal out-of-plane lattice expansion of about 1.7% has been reached for the $1\times10^{16}$ He cm$^{-2}$ dosed SnO$_2$ thin films with its structural quality being not deteriorated by low energy ion implantation[61]. However, when noble gas atoms are implanted at high fluence in covalent semiconductors, tiny noble gas bubbles are formed homogeneously distributed in a layer with its depth being controlled by the implantation energy. The implanted regions are partially amorphized due to the accumulation of ion damage. During thermal annealing using a conventional furnace, the bubbles evolve into larger bubbles[62]. Fortunately, it has been demonstrated very recently that the nanosecond laser pulse annealing can efficiently suppress the Ar bubbles to have good recrystallization of ion-implantation induced amorphous in sharp contrast to the conventional furnace thermal annealing[63]. Although many issues remain to be solved for experimental realization, we view our proposal positively.

In summary, we proposed a CMOS-compatible scheme to acquire the direct-bandgap emission from Ge using a strain-doping approach to expand the lattice to achieve the indirect-to-direct bandgap transition. Within this scheme, we have computationally demonstrated the direct-bandgap Ge doped with Li, He, Ne, Ar, and Kr, respectively. We show that, by increasing the doping concentration, the direct and fundamental bandgap can be tuned to cover the mid- to far-infrared spectral region. Feasibility of the approach has been discussed. Therefore, we provide a route toward silicon-based light emitters spanning a wide spectrum, including the 2 μm wavelength a leading candidate for the next-generation optical communication.

## Methods

### Basic Density Functional Theory calculation setup

All calculations were performed within the framework of density functional theory (DFT)[64–66], as implemented in Vienna Ab-initio Simulation Package (VASP). We used the projector augmented wave (PAW) pseudopotential formalism[67] with a plane wave cutoff of 500 eV. The Ge 3d electrons are treaded as valence electrons. An $8\times8\times8$ Γ-centered **k**-mesh (for primitive unit cell) or equivalent **k**-mesh (for supercells) is used for the Brillouin Zone sampling.

### Crystal model of the strain-doped Ge

For modeling the Ge:X (X = Li, He, Ne, Ar, and Kr) systems, we used the experimental lattice constant for crystalline Ge at 300 K[46]. For strain-doped Ge, where experimental data is unavailable, we used the DFT-predicted lattice constants multiplied by a constant c. This constant c represents the ratio between the experimental lattice constant of crystalline Ge, and the DFT value for crystalline Ge (i.e., $c = a_{expl}(Ge)/a_{DFT}(Ge)$). Structural relaxations were performed using the Perdew–Burke–Ernzerhof (PBE) exchange–correlation functional[68,69] in a two-step manner: (1) deriving the equilibrium lattice constant by fitting the energy-lattice parameter curve; (2) fixing the lattice to the equilibrium lattice constant and relaxing

the atomic positions until the change of the total energy is smaller than $10^{-5}$ eV. We note that because the lattice parameter and atomic positions are usually coupled. In these impurity inserted Ge systems, not relaxing lattice and atomic position simultaneously may introduce some excess strain for certain given impurity concentration. We envision a completely relaxed structure may results in some changes in the values of the direct and indirect bandgaps, but this should not affect the main trend we have demonstrated in this paper. Because the excitation spectra of Li in bulk Si and Ge[70] indicate that the Li impurity mostly occupies the interstitial positions with tetrahedral symmetry, subsequently confirmed by the electron paramagnetic resonance (EPR) experiment[71], we can safely assume that all inserted atoms distribute randomly but exclusively at the tetrahedral interstitial ($T_d$) sites in Ge. The assumption is further justified by earlier DFT results[72] and our calculated formation energy (See Supplementary Note 2 and Supplementary Table 1). We utilized the special quasi-random structures (SQS) method[47] to generate the atomic configurations of Ge:X for various atomic concentration of X, employing the "mcsqs" module[73] implemented in the ATAT package. The resulted supercells ranged from 32 atoms to 128 atoms per unit cell. Our DFT results comparing the SQS structure with a few different Li distribution configures of $Ge_{32}Li_2$ systems (See Supplementary Note 3 and Supplementary Fig. 2) show that they have very similar energies of the direct and indirect bandgaps, suggesting the role of Li distribution is subordinate.

### Electronic and optical properties calculations

We employed the hybrid functional (HSE06)[45] to calculate the electronic structure because PBE functional often tends to underestimate the bandgap of semiconductors. Because the random distribution of implanted atoms breaks the translational symmetry of the Ge crystal, causing energy states at different high-symmetry k points are folded into Γ and resulting in admixtures of these states. The direct bandgap $E_g^{\Gamma}$ and indirect bandgap $E_g^{L}$ of interstitial filled Ge were determined from the "unfolded" effective band structure[74]. The optical emission performance of the material is evaluated by the dipole matrix elements between the band-edges of the conduction and valence band $\boldsymbol{P} = i\langle c|\hat{\boldsymbol{p}}|v\rangle$. Its value is given in the energy unit following $E_P = 2\boldsymbol{P}^2/m_0$[48], where $m_0$ is the mass of an electron.

### Data availability

The results data used for the plotting of the Figures generated in this study have been deposited in the figshare database with the identifier [data https://doi.org/10.6084/m9.figshare.24457297][75]. The raw DFT data generated in this study are available upon restricted access for due to size considerations, access can be obtained by request from the corresponding author.

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

## Acknowledgements

The work was supported by National Key Research and Development Program of China under Grant No. 2018YFB2200105, the Key Research Program of Frontier Sciences, CAS under grant No. ZDBS-LY-JSC019, CAS Project for Young Scientists in Basic Research under grant

No. YSBR-026, the Strategic Priority Research Program of CAS under grant No. XDB43020000, and the National Natural Science Foundation of China (NSFC) under grant No. 11925407. The DFT calculations of this work were completed by L.D.Y. during his Ph.D. studies in J. W. Luo's group.

## Author contributions

J.W.L. and S.S.L. conceived and supervised the project. L.D.Y. and J.W.L. developed the idea and designed the research. L.D.Y. performed the density functional calculations and plot the Figures. L.D.Y., S.S.L., and J.W.L. analyzed the results and wrote the manuscript.

## Competing interests

The authors declare no competing interests.
