## [Peer Review File · Nature Communications]

REVIEWER COMMENTS

Reviewer #1 (Remarks to the Author):

The manuscript titled "Direct bandgap emission from strain-doped germanium" presents a theoretical exploration of a novel method to alter the bandgap nature of cubic Ge from indirect to direct by inducing strain through the implantation of Li and noble gas atoms. Although the manuscript acknowledges the usage of this technique in oxides and semiconductors like Si, GaP, and ZnP, its innovation lies in elucidating the underlying mechanism driving this shift—namely, strain induced by doping, which is a valuable, but not unexpected result. The manuscript goes on to provide specific doping concentrations required to achieve the desired transformation from indirect-to-direct bandgap for Li, He, Ne, Ar, and Kr. Optimism for the realization of the desired strain levels is drawn from an experimental study which used He-ion implantation to increase the lattice constant by 1% in the lateral direction.

Overall, the manuscript offers relevant information for a strain-induced indirect-to-direct bandgap transition in Ge. It examines the driving mechanism behind this transition and establishes the essential doping concentrations. However, information about the limitations of the mechanism is missing. Incorporating details about the upper limits of doping concentrations, the stability of the structures (dopant diffusion), and other possible issues for all desired dopants should be included. Investigating the impact of these (extremely high) doping levels on crystal structure disorder and the density of optically active defects, and the electronic transport properties should shed light on these limitations. Below I have provided some remarks on the text:

Remarks

- p.2 line 15: The authors are commenting on the feasibility of GeSn heterostructures. Along similar lines, there should be a more detailed discussion about the feasibility of noble gas doping of Ge in a CMOS compatible way.
- p.2 line: 26 "hexagonal Ge which was predicted to have a very weak optical dipole" Experimental studies have shown the opposite.
- Figure 1c: Misspelling: direct and indirect.
- P.3 line 33: "due to we adopted an experimental lattice constant instead of the DFT value used in the literature" it should be highlighted if otherwise theoretical or experimental values are used for the calculations.
- P.4 line 6: "is unlike to bond with Ge" please specify.
- P. 4 line 11: Is it correct that the same doping concentration is needed for Ar and Kr? And if so, can you elaborate why this is?
- P.4 line 33: "helium ion implantation to increase the out-of-plane lattice constant by 1%" was there ever a higher strain induced by noble gas doping? 2% is needed according to the calculations.
- P.5 line 1: The phrase "silicon-based" is misleading. Only Ge was discussed in this manuscript. 'Ge-based' should be used, since the mechanism for Si is described as "vastly different" in p.4 line 25.
- P.5 line1: The claim "proof-of-concept" is misleading. The theoretical groundwork is presented here, inviting for experimental proof-of-concept studies.
- Bandgap and band gap are both used simultaneously. Please choose one spelling.

Reviewer #2 (Remarks to the Author):

Using theoretical methods the authors describe the realization of an old idea that under tensile hydrostatic strain of a few percent the pure s-like conduction band minimum (CBM) at Gamma is shifted below the sp-like CBM at L making germanium (Ge) to a direct semiconductor with a strong optical transition strength. As origin of such a tensile strain they suggest to implant Li atoms on an interstitial site. The corresponding consequences are obtained by means of the density functional theory for atomic structure and energetics combined with a hybrid functional approach to the band structure. The authors describe a novel option to make bulk cubic Ge to an active light-emitting material and, therefore, a possibility to unify optoelectronics with Si electronics based on the CMOS technology.

I suggest publication of the manuscript with some modifications following the questions/comments below:

(i) I wonder that about 3% Li incorporation, i.e. 3% tensile strain, is sufficient. Other studies, even for hexagonal Ge, e.g. R. Suckert et al., PRMaterials 5, 024602 (2021), suggest larger strain values above 5%. Please comment.

(ii) The contradictory strain effect of a $\text{Si}_x\text{Ge}_{1-x}$ substrate discussed around Fig. S2 should be mentioned in the main text. However, Si-rich substrates as in Figs. S2 (d), (e) and (f) should be dropped down, since pseudomorphic growth of Ge is not possible for these substrates.

(iii) Why Li interstitials do not influence the electronic structure of the Ge host in the gap region? There are no Li-induced impurity levels?

(iv) How big are the slabs/supercells used for the modelling of the Li-doped Ge? Does the Li distribution play a role? Please mention the answers in the main text.

Reviewer #3 (Remarks to the Author):

This is a concise manuscript reporting band structure changes in Ge when the lattice is expanded by Li incorporation or rare gas incorporation. The band structure modifications of Ge by strain are already well known and established. It is also known from phase diagram studies that Li incorporation expands the lattice. The new result is that Li incorporation leads to similar band structure changes as lattice expansion without Li. This does not seem to be a very important result. Some specific comments follow:

There is not sufficient information in the methods description to understand the precision of the calculations. More details about the choices of computational parameters and justification of these is needed. This includes both the code parameters and choice of functionals.

The justification of assuming the tetrahedral site for Li is weak. It is noted that the energy of one of the interstitial sites is very similar. Does this mean that Li is expected to diffuse easily through the lattice? If so this may lead to a phase separation. The phase diagram contains Li rich ordered phases including one that appears at a relatively low temperature. This possibility is not discussed in the manuscript.

Li with its valence electron may be expected to alter the Fermi energy and serve as a dopant. It is not clear what practical use there would be for a direct gap version of Ge with heavy doping corresponding to multiple percent Li. The Fermi level shifts with Li and their implications are not discussed, which is a major flaw in the presentation.

The energetics of rare gas insertion are much less favorable than Li. This suggests that implantation may lead to other issues such as bubble formation and mechanical damage rather than uniform lattice expansion.

The arguments for compatibility with existing CMOS technology are weak. Current technology is not based on Ge but is based on strained Si (typically grown on layers of Si-Ge alloy to achieve strain, but not as the active layers). It is not clear at all why Li alloyed or rare gas implanted Ge would be compatible with CMOS technology.

The band gap of Ge is less than 0.7 eV, and importantly is significantly lower than Si. This makes the idea of strained Ge as replacement for an optical material compatible with Si less meaningful.

RESPONSE TO REVIEWERS' COMMENTS:

Reviewer #1 comments and author response

Reviewer#1 Comment 1: Overall, the manuscript offers relevant information for a strain-induced indirect-to-direct bandgap transition in Ge. It examines the driving mechanism behind this transition and establishes the essential doping concentrations. However, information about the limitations of the mechanism is missing. Incorporating details about the upper limits of doping concentrations, the stability of the structures (dopant diffusion), and other possible issues for all desired dopants should be included. Investigating the impact of these (extremely high) doping levels on crystal structure disorder and the density of optically active defects, and the electronic transport properties should shed light on these limitations.

Authors response: We appreciate the reviewer's positive feedback and acknowledge their valid concerns about the limitations of our proposed scheme. We have discussed the limitations of our proposed mechanism in the **Discuss** section, Page 5,6:

"Diffusion of Li into Ge followed by drift to compensate acceptor sites is a standard technology to fabricate the Ge:Li gamma-ray detectors developed in the 1960s. [Progress in Particle and Nuclear Physics 60, 283-337 (2008)] Although the solubility of Li in Ge crystal falls within the range of 10^{17} - 10^{18} cm⁻³[Journal of Physics and Chemistry of Solids 3, 74-76 (1957)], the Li concentration as high as 1×10^{21} cm⁻³ (~ 9.0 at.%) has been reported in the surface layer of Si_xGe_{1-x} (x=97.6 and 94.6%) crystals following a 10-minute diffusion at 320 °C measured directly by performing secondary mass ion spectrometry (SIMS). [Detectors and Associated Equipment 617, 588-590 (2010)] The results exhibit a clear decrease in Li concentration versus distance from the surface and higher Ge composition in Si_xGe_{1-x} crystals having a thicker high Li concentration surface layer with a higher peak concentration. It was attributed to the enhanced diffusion of lithium through interstitial positions by local strains created by Ge atoms. The observed low Li concentration in the Li-diffused crystalline Ge may exclude the possibility of Li-rich Li_xGe phases [Journal of Phase Equilibria 18, 289-294 (1997)], which occurs usually in the electrochemical lithiation of Ge [Physical Review B 90, 054111 (2014)]. To suppress the out-diffusion of Li atoms, we could deposit a highly compact oxide layer on top of the Ge:Li layer. The accumulation of Li atoms at the Ge surface and the wide bandgap of the deposition oxide layer may create a triangular-well confinement potential across the oxide/Ge:Li junction for both electrons and holes, which will benefit the lasing.

Doping far beyond the solubility limit can also be obtained by non-equilibrium doping techniques such as ion implantation followed by ultrafast annealing. [Skorupa, W. & Schmidt, H. Subsecond annealing of advanced materials: annealing by lasers, flash lamps and swift heavy ions. Vol. 192 (Springer Science & Business Media, 2013)] For instance, a saturation of interstitial lithium in the Si membrane at a concentration of about 10 at.% has been demonstrated by a direct-write lithiation of 35 nm thick crystalline Si membranes using a focused ion beam of Li⁺. [ACS Nano 13, 8012-8022 (2019)] The interstitial Li impurities introduce no deep level within the Ge bandgap, but acts as a donor in Ge. [Physical Review 91, 193-193 (1953)] Because Li has one

valence electron in its 2s orbit, it shifts the Fermi level into the conduction band due to a hyper-doping, which may limit the light-emitting performance...”

Below I have provided some remarks on the text:

Reviewer#1 Comment 2: *p.2 line 15: The authors are commenting on the feasibility of GeSn heterostructures. Along similar lines, there should be a more detailed discussion about the feasibility of noble gas doping of Ge in a CMOS compatible way.*

Authors response: We have discussed the feasibility of noble gas doping of Ge in a CMOS-compatible way in the **Discussion** section, page 6:

*“Unlike Li, noble gas atoms with closed-shell valence electron configuration are chemically inert and will not donate electrons to Ge; thus, more feasible for strain-doping induced indirect-to-direct bandgap transition in Ge. This is encouraged by a recent experiment [Physical Review Letters **114**, 256801 (2015)] that has successfully used helium ion implantation to increase the out-of-plane lattice constant by 1% in epitaxial $\text{La}_{0.7}\text{Sr}_{0.3}\text{MnO}_3$ thin films. A maximal out-of-plane lattice expansion of about 1.7% has been reached for the $10 \times 10^{15} \text{ He cm}^{-2}$ dosed SnO_2 thin films with its structural quality being not deteriorated by low energy ion implantation. [Nano letters **16**, 1782-1786 (2016)] However, when noble gas atoms are implanted at high fluence in covalent semiconductors, tiny noble gas bubbles are formed homogeneously distributed in a layer with its depth being controlled by the implantation energy. The implanted regions are partially amorphized due to the accumulation of ion damage. During thermal annealing using a conventional furnace, the bubbles evolve into larger bubbles. [Physical Review B **97**, 104102 (2018)] Fortunately, it has been demonstrated very recently that the nanosecond laser pulse annealing can efficiently suppress the Ar bubbles to have good recrystallization of ion-implantation induced amorphous in sharp contrast to the conventional furnace thermal annealing. [Journal of Applied Physics **133** (2023)] Although many issues remain to be solved for experimental realization, we view our proposal positively.”*

Reviewer#1 Comment 3: *p.2 line: 26 “hexagonal Ge which was predicted to have a very weak optical dipole” Experimental studies have shown the opposite.*

Authors response: We have added a brief comment on the experimental hexagonal Ge study [Nature 580, 205-209 (2020)] in the introduction:

“It sharply contrasts with the direct bandgap hexagonal Ge, which was predicted to exhibit a very weak optical dipole transition [Phys. Rev. Mat. 3, 034602 (2019)] due to its quasi-direct band gap³⁵. The recently observed finite emission in hexagonal Ge [Nature 580, 205-209 (2020)] may be a consequence of extrinsic effects [Phys. Rev. Mat. 5, 024602 (2021)].”

Reviewer#1 Comment 4: *Figure 1c: Misspelling: direct and indirect.*

Authors response: We have corrected it.

Reviewer#1 Comment 5: P.3 line 33: “due to we adopted an experimental lattice constant instead of the DFT value used in the literature” it should be highlighted if otherwise theoretical or experimental values are used for the calculations.

Authors response: Details of the DFT settings are included in the revised Method section. To highlight if otherwise theoretical or experimental values are used for the calculations, we have provided information for the basic setting of our calculations at the beginning of **Result** section page 3:

“We adopt the experimental lattice parameter for crystalline Ge at 300 K [Acta Crystallographica 15, 6, 578-582 (1962)], and scaled lattice constants for doped Ge systems where no experimental data is available.”

Reviewer#1 Comment 6: P.4 line 6: “is unlike to bond with Ge” please specify.

Authors response: We have rephrased this sentence in the last paragraph of the **Results** section, page 4:

“Unlike Li, noble gas atoms with closed-shell valence electron configuration are chemically inert and will not donate electrons to Ge; Thus, being an ideal dopant for strain doping.”

Reviewer#1 Comment 7: P. 4 line 11: Is it correct that the same doping concentration is needed for Ar and Kr? And if so, can you elaborate why this is?

Authors response: The critical doping concentration derived here may not precisely reflect the exact transition concentration, given the largest supercell considered is 128 atoms. For Ar and Kr, the transition occurs between 0 to 0.8% (corresponds to 0 to 1 atom in a 128 atoms supercell). The actual transition concentration, as evidenced by the crossing of E_g^Γ and E_g^L in Fig. 4(c) and (d), is expected to be smaller than 0.8%. A closer examination seems to suggest that the concentration required (cross point) for Kr should be even smaller than that of Ar, which aligns with the fact that Kr has larger atomic radius than Ar. We have clarified this in the last paragraph of the **Results** section, page 4,5:

“We should note that the predicted critical doping concentration may not be very precise given the largest supercell considered here contains only 128 atoms and the critical concentration 0.8 at.% corresponds to 1 atom in a 128 atoms-supercell...”

Reviewer#1 Comment 8: P.4 line 33: “helium ion implantation to increase the out-of-plane lattice constant by 1%” was there ever a higher strain induced by noble gas doping? 2% is needed according to the calculations.

Authors response: The controlled realization of 1% out-of-plane lattice expansion [Physical Review Letters 114, 256801 (2015)] is encouraging. It suggests the helium ion implantation is a potential feasible scheme. A maximal out-of-plane lattice expansion of about 1.7% has been reached for the 10×10^{15} He cm⁻² dosed SnO₂ thin films with its structural quality being not

deteriorated by low energy ion implantation. [Nano letters 16, 1782-1786 (2016)] The current manuscript examines the proposed scheme with DFT provides a theoretical groundwork, experimentally achieving 2% tensile strain are challenging but promising. See also our reply to comment 2.

Reviewer#1 Comment 9: *P.5 line 1: The phrase “silicon-based” is misleading. Only Ge was discussed in this manuscript. ‘Ge-based’ should be used, since the mechanism for Si is described as “vastly different” in p.4 line 25.*

Authors response: We have changed “silicon-based” to “Ge-based”.

Reviewer#1 Comment 10: *P.5 line1: The claim “proof-of-concept” is misleading. The theoretical groundwork is presented here, inviting for experimental proof-of-concept studies.*

Authors response: We have removed the misleading claim and added at the end of the **Abstract**, **Page 1:**

“This novel Ge-based light-emitting approach presents exciting prospects for surpassing the physical limitations of silicon technology in the field of photonics calling for experimental proof-of-concept studies.”

Reviewer#1 Comment 11: *Bandgap and band gap are both used simultaneously. Please choose one spelling.*

Authors response: We now used “bandgap” in all cases.

Reviewer #2 comments and author response

Reviewer#2 Comment 1: *Using theoretical methods the authors describe the realization of an old idea that under tensile hydrostatic strain of a few percent the pure s-like conduction band minimum (CBM) at Gamma is shifted below the sp-like CBM at L making germanium (Ge) to a direct semiconductor with a strong optical transition strength. As origin of such a tensile strain they suggest to implant Li atoms on an interstitial site. The corresponding consequences are obtained by means of the density functional theory for atomic structure and energetics combined with a hybrid functional approach to the band structure. The authors describe a novel option to make bulk cubic Ge to an active light-emitting material and, therefore, a possibility to unify optoelectronics with Si electronics based on the CMOS technology.*

Authors response: Thank you.

Reviewer#2 Comment 2: *I wonder that about 3% Li incorporation, i.e. 3% tensile strain, is sufficient. Other studies, even for hexagonal Ge, e.g. R. Suckert et al., PRMaterials 5, 024602 (2021), suggest larger strain values above 5%. Please comment.*

Authors response: We have cited this paper and compared it to our study in the **Results** section, page 3,4:

“Note that the L-valley of cubic Ge folds into the Γ -point in the hexagonal Ge and couples with the high-lying state derived from the Γ -valley of cubic Ge. This coupling repels the Γ -valley-derived state upward and L-valley-derived state downward, necessitating a larger hydrostatic tensile strain of approximately 5% to transform the quasi-direct bandgap into a direct bandgap in hexagonal Ge [Phys. Rev. Mat. 5, 024602 (2021)].”

Reviewer#2 Comment 3: *The contradictory strain effect of a SixGe_{1-x} substrate discussed around Fig. S2 should be mentioned in the main text. However, Si-rich substrates as in Figs. S2 (d), (e) and (f) should be dropped down, since pseudomorphic growth of Ge is not possible for these substrates.*

Authors response: Fig. S2 show the SixGe_{1-x} substrate has a reverse effect on the direct and indirect bandgap reversion. This is because the lattice constant of Ge is larger than that of Si, epitaxial Ge layers and Ge nanostructures on Si are usually compressively strained. The compressive strain effect shown in Fig. S2 does not contradict (actually, agrees) the tensile strain effect shown in Fig. 2. The adverse impact of the SixGe_{1-x} substrate can be mitigated when epitaxial Ge films are thicker and grow at high temperature using a two-step growth method: (1) A Ge layer thicker than 200 nm can nearly completely relax at growth temperatures above 600 °C; (2) when cooled to room temperature, tensile strain, instead of compressive strain, can accumulate in the Ge layers because Ge exhibits larger thermal expansion coefficient compared to Si. Indeed, thermally induced tensile strain of 0.25% has been achieved using this approach. [Nature Photonics

4, 527 (2010)]. Relaxed GeSn buffer layer on Si can be used to further enhance the tensile strain in Ge. [Appl. Phys. Lett. 90, 061915 (2007); Chem. Mater. 19,5910-5925 (2007)]

To clarify the above points, we have moved previous Fig. S2 to the results section (now Fig. 3), and have discussed the biaxial compressive strain effect induced by the substrate, page 4:

“So far, we have demonstrated the indirect-to-direct bandgap transition by incorporating Li atoms into free-standing Ge. We now consider the effect of biaxial strain on Ge:Li systems arising from a $\text{Si}_x\text{Ge}_{1-x}$ substrate. Figure 3 shows the evolution of energy levels of the direct and indirect bandgaps E_g^Γ and E_g^L as a function of Li concentration in the Ge:Li system epitaxially grown on the $\text{Si}_x\text{Ge}_{1-x}$ substrates ($x=0, 0.2, 0.4$). In comparison to the freestanding case, the pseudomorphous Ge:Li layer on a Ge substrate ($x=0$) requires a larger Li concentration (6.2%) to achieve a direct bandgap. This biaxial strain effect induced by the $\text{Si}_x\text{Ge}_{1-x}$ substrate becomes more pronounced with a lower Ge composition. Fig. 3c shows that when $x=0.4$ the increase of Li concentration is unlikely to reverse the order of E_g^Γ and E_g^L , and thus cannot transform the Ge:Li from indirect bandgap to direct bandgap. It implies that a Ge-rich $\text{Si}_x\text{Ge}_{1-x}$ substrate (<0.4) is necessary to obtain Li-doped direct bandgap Ge. Fortunately, the technology of high-quality relaxed Ge layers grown on the Si substrate is now mature. [APL Photonics 7 (2022), Nature Photonics 15, 925-931 (2021), Nature communications 13, 6624 (2022)]”

Figure 3 | Effect of biaxial compressive strain induced by $\text{Si}_x\text{Ge}_{1-x}$ substrate. Direct bandgap E_g^Γ and indirect bandgap E_g^L as a function of Li atomic concentration in Ge:Li system on $\text{Si}_x\text{Ge}_{1-x}$ substrate for (a) $x=0$; (b) $x=0.2$; and (c) $x=0.4$.

Reviewer#2 Comment 4: *Why Li interstitials do not influence the electronic structure of the Ge host in the gap region? There are no Li-induced impurity levels?*

Authors response: We have briefly commented this in the **Discussion** section, Page 5,6:

“The interstitial Li impurities introduce no deep level within the Ge bandgap, but acts as a donor in Ge. [S. Fuller and J. A. Ditzenberger, Phys. Rev. 91, 193 (1953) F. J. Morin, J. P. Maita, R. G. Shulman, and N. B Hannay, Phys. Rev. 96, 833 (1953)] Because Li has one valence electron in its 2s orbit, it shifts the Fermi level into the conduction band ...”

Reviewer#2 Comment 5: *How big are the slabs/supercells used for the modelling of the Li-doped Ge? Does the Li distribution play a role? Please mention the answers in the main text.*

Authors response: We used 32 and 64 atoms supercells for the modeling of the Li-doped Ge. To answer the second question, we have added a new Figure (Figure S2) in the Supplementary Information comparing the ensuring changes of the direct and indirect bandgap among the SQS structure and other five random configurations of the same Li concentrations ($\text{Ge}_{32}\text{Li}_2$). Our DFT results show the main effect of Li insertion in Ge is induce tensile strain, the role of the Li distribution is subordinate. We have commented this in the main text **Method** section, page 7:

“Our DFT results comparing the SQS structure with a few different Li distribution configures of $\text{Ge}_{32}\text{Li}_2$ systems (see Supplementary Information Figure S2) show that they have very similar energies of the direct and indirect bandgaps, suggesting the role of Li distribution is subordinate.”

$\text{Ge}_{32}\text{Li}_2$ (configurations)	Crystal structure	E_g^T (eV)	E_g^L (eV)	$E_P = 2P^2/m$ (eV)
SQS		0.407	0.515	18.569
Random configuration 1		0.404	0.502	16.006
Random configuration 2		0.408	0.497	16.409
Random configuration 3		0.404	0.502	16.006
Random configuration 4		0.423	0.505	18.858
Random configuration 5		0.425	0.490	19.004

Figure S2 | Role of Li distribution. DFT calculated direct bandgap, indirect bandgap and the corresponding fundamental bandgap dipole matrix element for the SQS configure (used to obtain result of Fig. 1) and other five random configurations of in $\text{Ge}_{32}\text{Li}_2$ (6.25% at. Li in Ge).

Reviewer #3 comments and author response

Reviewer#3 Comment 1: *This is a concise manuscript reporting band structure changes in Ge when the lattice is expanded by Li incorporation or rare gas incorporation. The band structure modifications of Ge by strain are already well known and established. It is also known from phase diagram studies that Li incorporation expands the lattice. The new result is that Li incorporation leads to similar band structure changes as lattice expansion without Li. This does not seem to be a very important result.*

Author response: The doubts the referee had regarding the importance of this work, or more generally the scheme of using Ge for Si photonics, is mainly based on the judgements (as suggested in Review#3 comment 7 and 8 below) that (1) the bandgap of strained Ge is less than 0.7 eV much smaller than that of Si, therefore not very useful; (2) Ge-based materials may not be completely compatible with the CMOS technology. Respectfully, we have a different perspective.

First, the addition of Ge to the Si photonics platform has been the key enabler for the manufacturing of active optical devices [APL Photon. 7, 050901 (2022)]. Specifically, thanks to its almost direct bandgap, Ge has long been pursued as an optical material compatible with Si. A bandgap of 0.7 eV or less allows the realization of the Si photonics operating at near-infrared (NIR) wavelengths. This is a leading candidate for the next optical communication window. [Nature Photonics 9, 393-396 (2015); Nature Photonics 16, 744-745 (2022); Nature Photonics 9, 358-359 (2015)]

Second, the Ge integration in Si has steadily evolved over the years. Ge has been used for the realization of Si-integrated waveguide-photodetectors [Nature Photonics 4, 527 (2010), Nature Photonics 15, 925 (2021)]. Ge is now recognized as a “standard” CMOS material, by institutes and foundries (such as IHP, CEA-Leti, IMEC, and AMF, to name a few) who offers electronic-photonics integrated circuits (ePIC) and process design kits (PDKs). [APL Photon. 7, 050901 (2022)] The complete compatibility of Ge with the CMOS technology shall be within reach.

The significance of this work is mainly that we offer a new concept to realize a direct bandgap Ge-based material. This material platform is promising for building monolithic lasers and overcome the last big challenge for Si photonics. We have addressed every comment and criticism the referee raised in detail below.

Some specific comments follow:

Reviewer#3 Comment 2: *There is not sufficient information in the methods description to understand the precision of the calculations. More details about the choices of computational parameters and justification of these is needed. This includes both the code parameters and choice of functionals.*

Author response: We have reworked the **Method** Section to include more computational details, page 6,7.

Reviewer#3 Comment 3: *The justification of assuming the tetrahedral site for Li is weak.*

Author response: It is well established that, in bulk Ge, Li impurity prefers to occupy an interstitial site as a shallow donor rather than a substitutional dopant. To justify this, we have revised the text the **Method** section, page 7:

“...Because the excitation spectra of Li in bulk Si and Ge[Phys. Rev. 138, A882(1965)] indicate that the Li impurity mostly occupies the interstitial positions with tetrahedral symmetry, subsequently confirmed by the electron paramagnetic resonance (EPR) experiment[Phys.Rev.B 1, 4071(1970)], we can safely assume that all inserted atoms distribute randomly but exclusively at the tetrahedral interstitial (Td) sites in Ge. The assumption is further justified by earlier DFT results [J. Phys.: Condens. Matter 22, 415501 (2010)], and our calculated formation energy (see Supplementary Information section A).”

Reviewer#3 Comment 4: *It is noted that the energy of one of the interstitial sites is very similar. Does this mean that Li is expected to diffuse easily through the lattice? If so this may lead to a phase separation. The phase diagram contains Li rich ordered phases including one that appears at a relatively low temperature. This possibility is not discussed in the manuscript.*

Author response: Yes, Li diffuses easily through the lattice with a significantly large diffusion coefficient of Li in Si and Ge measured experimentally [Phys. Rev. 91 (1953) 193; J. Solid State Chem. 37 (3) (1981) 271]. We have discussed the issue of Li diffusion in the **Discussion** section, Page 5:

“... there are a few realistic issues associated with Li. First, while the required doping atomic concentration is in the order of a few percent ($\sim 10^{20} \text{ cm}^{-3}$), the solubility of the investigated dopants is much lower. The solubility of Li in Ge fails within the range of 10^{17} - 10^{18} cm^{-3} , with a maximum atomic concentration reaching $(1.7 \pm 0.5) \times 10^{-4}$ at around 800°C. [Journal of Physics and Chemistry of solids 3, 74-76 (1957)] Second, thermal oxidation, a key process of CMOS, can lead to the formation of unwanted complexes of Li and oxygen. [Journal of Phase Equilibria Vol. 18 No. 3 1997] Third, Li is chemically reactive, and there are multiple intermediate phases exist in the Li-Si phase diagram [Journal of Solid State Chem. 37, 271–278 (1981)] and Li–Ge phase diagram [Journal of Phase Equilibria 18, 289-294 (1997)]. Due to Li’s rapid diffusion in Ge [Phys. Rev. 91 (1953) 193; J. Solid State Chem. 37 (3) (1981) 271], Li may accumulate at the Ge surface, leading to the formation of segregated Li-rich ordered phases. [Chem Mater 30,10,3254-3264 (2018)] Last but not least, Li acts as a shallow donor in Ge, donating its 2s electron to Ge [S. Fuller and J. A. Ditzenberger, Phys. Rev. 91, 193 (1953) F. J. Morin, J. P. Maita, R. G. Shulman, and N. B Hannay, Phys. Rev. 96, 833 (1953)] While it does not create an additional level within the bandgap, it shifts the Fermi level of Ge into its conduction band. This often results in plasma limiting the light-emitting performance. Considering these drawbacks of Li, we alternatively suggest using noble gas atoms to expand the Ge lattice.”

We suggest noble gas atom implantation might be an alternative

Reviewer#3 Comment 5: *Li with its valence electron may be expected to alter the Fermi energy and serve as a dopant. It is not clear what practical use there would be for a direct gap version of Ge with heavy doping corresponding to multiple percent Li. The Fermi level shifts with Li and their implications are not discussed, which is a major flaw in the presentation.*

Author response: We have discussed the issue of Li diffusion in the Results section and suggested noble gas atom implantation might be an alternative in the first paragraph of the **Discussion** section, Page 5,6:

“The interstitial Li impurities introduce no deep level within the Ge bandgap, but acts as a donor in Ge. [S. Fuller and J. A. Ditzenberger, Phys. Rev. 91, 193 (1953) F. J. Morin, J. P. Maita, R. G. Shulman, and N. B Hannay, Phys. Rev. 96, 833 (1953)] Because Li has one valence electron in its 2s orbit, it shifts the Fermi level into the conduction band due to a hyper-doping, which may limit the light-emitting performance.”

Reviewer#3 Comment 6: *The energetics of rare gas insertion are much less favorable than Li. This suggests that implantation may lead to other issues such as bubble formation and mechanical damage rather than uniform lattice expansion.*

Authors response: We note bubbles can be desorbed at high temperature. We have discussed this for noble gas atom implantation in the **Discussion** section, page 7:

“...when noble gas atoms are implanted at high fluence in covalent semiconductors, tiny noble gas bubbles are formed homogeneously distributed in a layer with its depth being controlled by the implantation energy. The implanted regions are partially amorphized due to the accumulation of ion damage. During thermal annealing using a conventional furnace, the bubbles evolve into larger bubbles. [Physical Review B 97, 104102 (2018)] Fortunately, it has been demonstrated very recently that the nanosecond laser pulse annealing can efficiently suppress the Ar bubbles to have good recrystallization of ion-implantation induced amorphous in sharp contrast to the conventional furnace thermal annealing. [Journal of Applied Physics 133 (2023)] Although many issues remain to be solved for experimental realization, we view our proposal positively.”

Reviewer#3 Comment 7: *The band gap of Ge is less than 0.7 eV, and importantly is significantly lower than Si. This makes the idea of strained Ge as replacement for an optical material compatible with Si less meaningful.*

Authors response: We share a different perspective on this. The addition of Ge to the Si photonics platform has been the key enabler for the manufacturing of active optical devices [APL Photon. 7, 050901 (2022)]. Specifically, thanks to its almost direct bandgap, Ge has long been pursuit as an optical material compatible with Si. To emphasis the importance of Ge on Si technology, we have added a few sentences in the Introduction section, page 2:

“After the first successful introduction of Ge in the source/drain regions in the 90 nm logic technology node to create a uniaxial compressive strain on the Si channel [S. Thompson et al., IEDM 61-64 (2002)], the integration of Ge in Si has steadily evolved, and Ge is now recognized as a “standard” CMOS material [APL Photonics 7 (2022)]. Specifically, Ge has been demonstrated to realize the ultrafast (a 3 dB bandwidth of 265 GHz) Si-integrated waveguide-photodetectors in compatibility with silicon photonics and CMOS fabrication. [Nature Photonics 4, 527-534 (2010), Nature Photonics 15, 925-931 (2021)] Thanks to Ge’s almost direct bandgap--the direct bandgap E_g^F is only about 0.14 eV above its fundamental indirect bandgap E_g^I ...”

A bandgap of 0.7 eV or less allows the realization of the Si photonics operating at near-infrared (NIR) wavelengths. This is a leading candidate for the next optical communication window. [Nature Photonics 9, 393-396 (2015); Nature Photonics 16, 744-745 (2022); Nature Photonics 9, 358-359 (2015)] We have added a comment in the last paragraph of the **Results** section, page 5:

“...all systems can be engineered to direct-bandgap emission of the 2 μ m (0.62 eV) wavelength, which is a leading candidate for the next optical communication window. [Nature Photonics 9, 393-396 (2015); Nature Photonics 16, 744-745 (2022); Nature Photonics 9, 358-359 (2015)]”

Reviewer#3 Comment 8: *The arguments for compatibility with existing CMOS technology are weak. Current technology is not based on Ge but is based on strained Si (typically grown on layers of Si-Ge alloy to achieve strain, but not as the active layers). It is not clear at all why Li alloyed or rare gas implanted Ge would be compatible with CMOS technology.*

Authors response: Si has been the transistor channel material of choice throughout all CMOS technology generations. Up until the 7nm node, Ge has only been used as SiGe alloy in source and drain terminals to generate a uniaxial compressive strain to the Si channel. Moving forward, the existing Si-based technology is now benefiting from a shift towards Ge. TSMC’s 5nm technology is the first advanced logic production technology featuring SiGe as the channel material for p-type FinFET (<https://research.tsmc.com/english/research/logic/high-mobility-channel/publish-time-1.html>). TSMC reported the Si₂₀Ge₈₀ channel capped by a 1nm thick Si layer in the 3nm FINFET node (Mark Liu, “Unleashing the Future of Innovation,” 2021 IEEE International Solid-State Circuits Conference (ISSCC), Plenary Session 1.1, 2021.) On the other hand, after the first successful integration and commercialization of “Ge-inside” active optical cables (AOC) from Luxtera in 2007, the Ge integration in Si has steadily evolved, and now Ge is recognized as a “standard” CMOS material, with institutes and foundries (such as IHP, CEA-Leti, IMEC, and AMF, to name a few) offering electronic-photonics integrated circuits (ePIC) and process design kits (PDKs) [APL Photon. 7, 050901 (2022)]. The complete compatibility of Ge with the current CMOS technology shall be within reach soon.

We have discussed specifically the feasibility of noble gas doping of Ge in a CMOS-compatible way in the **Discussion** section, page 7:

*“Unlike Li, noble gas atoms with closed-shell valence electron configuration are chemically inert and will not donate electrons to Ge; thus, more feasible for strain-doping induced indirect-to-direct bandgap transition in Ge. This is encouraged by a recent experiment [Physical Review Letters **114**, 256801 (2015)] that has successfully used helium ion implantation to increase the out-*

*of-plane lattice constant by 1% in epitaxial $\text{La}_{0.7}\text{Sr}_{0.3}\text{MnO}_3$ thin films. A maximal out-of-plane lattice expansion of about 1.7% has been reached for the $10 \times 10^{15} \text{ He cm}^{-2}$ dosed SnO_2 thin films with its structural quality being not deteriorated by low energy ion implantation. [Nano letters **16**, 1782-1786 (2016)] However, when noble gas atoms are implanted at high fluence in covalent semiconductors, tiny noble gas bubbles are formed homogeneously distributed in a layer with its depth being controlled by the implantation energy. The implanted regions are partially amorphized due to the accumulation of ion damage. During thermal annealing using a conventional furnace, the bubbles evolve into larger bubbles. [Physical Review B **97**, 104102 (2018)] Fortunately, it has been demonstrated very recently that the nanosecond laser pulse annealing can efficiently suppress the Ar bubbles to have good recrystallization of ion-implantation induced amorphous in sharp contrast to the conventional furnace thermal annealing. [Journal of Applied Physics **133** (2023)] Although many issues remain to be solved for experimental realization, we view our proposal positively.”*

REVIEWER COMMENTS

Reviewer #1 (Remarks to the Author):

the comments and questions raised by the reviewers have been addressed seriously. Some points in the text remain controversial, but these are now at least explicitly mentioned. It remains an interesting idea, which may trigger experimental groups.

Reviewer #2 (Remarks to the Author):

The authors answered to all my questions satisfactorily and followed my suggestions, e.g. to move one figure from the Supplementary to the main text. Therefore, in principle, I recommend publication of the manuscript.

Personally, I ask the authors for minor supplementations:

- (i) Explain the meaning and the computation of E_p in the Method section.
- (ii) Argue, why highly n-doped (here with Li) Ge is still a material applicable for active optoelectronic purposes.

Reviewer #3 (Remarks to the Author):

The authors have responded to the reviewer comments. However, these responses are not adequate in all cases, and I do not recommend publication of this work.

They have added details about the calculational procedure, which is helpful. However, the statement that they used a two step procedure for relaxations, first relaxing the lattice parameter and then the atomic positions is not adequate. These are coupled and not relaxing them simultaneously could introduce significant error, for example finding excess strain for a given impurity concentration.

The statements about compatibility with CMOS are not reasonable. First of all, they seem to agree that Li diffuses. Adding an element that diffuses like Li to a CMOS process seems very unlikely to succeed. Generally, Na and especially K are to be avoided in the processes. The statement that rare gas implantation at high levels is fine in such processes also seems unsupported. In particular, the resulting damage is unlikely to be acceptable, and the idea of annealing it out seems unreasonable as this would remove the rare gas atoms and the strain.

RESPONSE TO REVIEWERS' COMMENTS:

Reviewer #1 comments and author response

Reviewer#1 Comment 1: Reviewer #1 (Remarks to the Author): the comments and questions raised by the reviewers have been addressed seriously. Some points in the text remain controversial, but these are now at least explicitly mentioned. It remains an interesting idea, which may trigger experimental groups.

Authors response: Thank you.

Reviewer #2 comments and author response

Reviewer#2 Comment 1: Reviewer #2 (Remarks to the Author): The authors answered to all my questions satisfactorily and followed my suggestions, e.g. to move one figure from the Supplementary to the main text. Therefore, in principle, I recommend publication of the manuscript. Personally, I ask the authors for minor supplementations: (i) Explain the meaning and the computation of E_p in the Method section. (ii) Argue, why highly n-doped (here with Li) Ge is still a material applicable for active optoelectronic purposes.

Authors response: For (i), E_p is the matrix element units in energy, which has been extensively reported for semiconductors. We have now explained it in the Method section:

“The optical emission performance of the material is evaluated by the dipole matrix elements between the band-edges of the conduction and valence bands $P = i\langle c | \hat{p} | v \rangle$. Its value is given in the energy unit following $E_p = 2P^2/m_0$ [I. Vurgaftman et al., J. Appl. Phys., 89, 5815 (2001)], where m_0 is the mass of an electron.”

For (ii), we agree with the reviewer that highly doped semiconductors may be unsuitable for optoelectronic devices mainly due to the absorption of light by free carriers in semiconductors, which results in optical loss for all photon wavelengths. Fortunately, we can still assess the transition from indirect to direct bandgap in Li-doped Ge from the photoluminescence spectroscopy since photo-excited carriers will relax to band edges in picosecond timescale and then radiatively recombine to generate photons. (see schematic illustration below) In our previous revision, we have included an explicit discussion on the limitations of highly n-doped Ge by Li and suggested it is more promising to use noble gas dopants for active optoelectronic device applications. To make it clearer, we have added an argument in the discussion section on page 6:

“Fortunately, we can still assess the transition from indirect to direct bandgap in Li-doped Ge from the photoluminescence spectroscopy since photo-excited carriers will relax to band edges in ps timescale and then radiatively recombine to generate photons. Therefore, it is still worth checking Li doped Ge experimentally, given that Li doping in Si and Ge is very accessible and has been widely applied in Li battery applications.”

Reviewer #3 comments and author response

Reviewer#3 Comment 1: Reviewer #3 (Remarks to the Author): They have added details about the calculational procedure, which is helpful. However, the statement that they used a two step procedure for relaxations, first relaxing the lattice parameter and then the atomic positions is not adequate. These are coupled and not relaxing them simultaneously could introduce significant error, for example finding excess strain for a give impurity concentration.

Authors response: We agree with the Reviewer that a complete relaxation of the structure requires simultaneously relax both the cell shape and atomic positions. A “fully” relaxed structure may result in some small changes in the values of the direct and indirect bandgaps but should not alter our conclusions. To make the analysis simple, we adopted a two-step procedure to keep the cell shape unchanged, such that we can easily project the electronic states onto the primitive diamond lattice and determine the direct bandgap E_g^Γ and indirect bandgap E_g^L for the interstitial filled Ge systems. We have included a discussion on the limitations of the current relaxation approach in the **Method** section.

“We note that because the lattice parameter and atomic positions are usually coupled in these impurity inserted Ge systems, not relaxing lattice and atomic position simultaneously may introduce some excess strain for certain given impurity concentration. We envision a completely relaxed structure may results in some changes in the value of the direct and indirect bandgaps, but this should not affect the main trend we have demonstrated in this paper.”

Reviewer#3 Comment 2: The statements about compatibility with CMOS are not reasonable. First of all, they seem to agree that Li diffuses. Adding an element that diffuses like Li to a CMOS process seems very unlikely to succeed. Generally, Na and especially are to be avoided in the processes. The statement that rare gas implantation at high levels is fine in such processes also seems unsupported. In particular, the resulting damage is unlikely to be acceptable, and the idea of annealing it out seems unreasonable as this would remove the rare gas atoms and the strain.

Authors response: We appreciate all the serious criticisms the reviewer had regarding the potential issues that might occur while implementing the proposed scheme. We agree that there are many challenges remain to be resolved for doping a few atomic percent of Li or noble gas atoms in Ge. We note that by using femtoseconds (rather than nanoseconds) ultrafast annealing technologies, we can locally repair the ion-implantation-caused damages without losing the rare gas atoms and the strain. Increasing the atom size from He, Ne, Ar, to Xe could reduce the diffusion. We also note the recent advancement in Ge-based technology provides reasonable insights on how these issues might be resolved in the near future. Both points are reflected in our previous revision where we have included an extensive discussion on the potential limitations and potential solutions.

It appears premature to draw definitive conclusions in contemplating the viability of the suggested scheme as either an unrealistic proposal or a potential avenue for on-chip light sources compatible with CMOS technology. Eventually, engaging experimental is the key to ascertain its potential. The proposal outlined in this paper represents a prospect that beckons further experimental explorations. This is specifically pointed out by referee #1: *“it remains an interesting idea, which may trigger experimental groups.”*

For these reasons, we believe our theoretical proposal is valid and has its own merit for publishing, this has been supported by Reviewer #1 and #2.

REVIEWERS' COMMENTS

Reviewer #2 (Remarks to the Author):

I suggest publication of the manuscript.